# Monitoring Arctic Permafrost – Examining the Contribution of Volunteered Geographic Information to Mapping Ice-Wedge Polygons

Pauline Walz<sup>1</sup>, Oliver Fritz<sup>1</sup>, Sabrina Marx<sup>1</sup>, Marlin M. Mueller<sup>2</sup>, Christian Thiel<sup>2</sup>, Josefine Lenz<sup>3</sup>, Soraya Kaiser<sup>3</sup>, Roxanne Frappier<sup>5</sup>, Alexander Zipf<sup>1,4</sup>, and Moritz Langer<sup>5,3</sup>

Abstract. This study evaluates the potential of Volunteered Geographic Information (VGI) for mapping and monitoring ice-wedge polygons in Arctic permafrost regions through two case studies in Alaska and Canada. We developed and tested a web-based mapping application that enables volunteers to identify ice-wedge polygon centroids in high-resolution aerial imagery, with data collected from 105 contributors as part of organized mapping events. The volunteer-contributed data achieved completeness scores of 88.74 % and 70.81 % for the Cape Blossom (Alaska) and Blueberry Hills (Canada) study regions respectively, with median positional accuracies of 1.29 m and 1.38 m (both validated against expert mapping data). Analysis shows that contributions from approximately five volunteers per polygon are sufficient to achieve reliable results. Using Voronoi diagrams derived from the crowd-sourced centroids, we successfully reconstructed ice-wedge polygon networks and extracted key geomorphological and hydrological parameters including polygon area, perimeter, and network topology. The results demonstrate that VGI can effectively support permafrost monitoring by enabling efficient mapping of ice-wedge polygons across large areas while maintaining high data quality standards.

<sup>&</sup>lt;sup>1</sup>Heidelberg Institute for Geoinformation Technology (HeiGIT), Schloss-Wolfsbrunnenweg 33, 69118 Heidelberg, Germany <sup>2</sup>German Aerospace Center (DLR), Institute of Data Science, Mälzerstraße 3-5, 07745 Jena, Germany

<sup>&</sup>lt;sup>3</sup>Permafrost Research Section, Alfred Wegener Institute Helmholtz Centre for Polar and Marine Research, Telegrafenberg A45, 14473 Potsdam, Germany

<sup>&</sup>lt;sup>4</sup>GIScience Research Group, Institute of Geography, Heidelberg University, Im Neuenheimer Feld 368, 69120 Heidelberg, Germany

<sup>&</sup>lt;sup>5</sup>Department of Earth Sciences, Vrije Universiteit Amsterdam, De Boelelaan 1085, 1081 HV Amsterdam, The Netherlands **Correspondence:** Oliver Fritz (oliver.fritz@heigit.org) and Moritz Langer (moritz.langer@awi.de)

#### 1 Introduction

Permafrost, the largest non-seasonal component of the cryosphere in area, plays a crucial role in Arctic ecosystems. Large-scale monitoring of its state and changes is urgently needed to better understand the direct impacts of climate warming in the Arctic, which vary greatly by region and are already in full swing (Nitzbon et al., 2024). As a subsurface thermal phenomenon, permafrost cannot be directly observed using remote sensing methods (Westermann et al., 2015). Instead, geomorphological structures of the Earth's surface, such as polygonal tundra, are used to identify and classify permafrost areas that are highly vulnerable to climate warming (e.g. Nitze et al., 2018; Runge et al., 2022). Due to the relatively small size of these polygonal land surface structures, very high-resolution image data is required for detection (Rettelbach et al., 2021). The rapid technological development of satellites has given rise to a growing database of high-resolution images available for the identification of these key indicators of permafrost and its condition.

The surface structures of ice-wedge polygons provide information about the presence of ice-rich permafrost (e.g. Bernard-Grand'Maison and Pollard, 2018). Their size and shape also provides information about the local climate and soil conditions that have controlled their development (Lachenbruch, 1962). In addition, the reconstruction of ice-wedge networks allows the drainability and hydrological changes of tundra ecosystems to be better delineated (Liljedahl et al., 2024). Thus, mapping the structure of ice-wedge polygonal networks is crucial for assessing the vulnerability of permafrost regions to climate change. These networks play a key role in controlling shifts in geomorphological and hydrological regimes as permafrost thaws and polygonal tundra degrades (Liljedahl et al., 2016; Nitzbon et al., 2019).

Several methods for detecting polygonal land surface structures have been developed and tested in the past, ranging from manual (e.g. Frappier and Lacelle, 2021) to semi-automated (e.g. Alexei N. Skurikhin and Wilson, 2013) detection techniques. More recently, the application of AI image analysis methods has substantially improved the ability to recognize and classify these structures (e.g. Zhang et al., 2018; Witharana et al., 2020). However, as a basis for training or as baseline data for verification, ground truth data is essential. This data is usually obtained through terrestrial surveying or visual interpretation of very high resolution imagery by experts. The generation of such baseline data is, however, labor intensive and therefore limited to a few regions that do not cover the full variability of land surface conditions under which polygonal structures occur. This limitation leads to substantial uncertainties in automatic recognition methods, which cannot reveal the subtle changes in polygonal structures that result when permafrost thaws (e.g. Witharana et al., 2021; Lousada et al., 2018). To improve and expand ground-truth data on ice-wedge polygons, we investigate the quality of geographic data generated by volunteers using a crowd-mapping approach that allows large areas to be mapped, utilizing human skills of structure recognition and context interpretation.

A considerable number of studies have demonstrated the potential and fitness for purpose of Volunteered Geographic Information (VGI) for a range of use cases, such as disaster response (Goodchild and Glennon, 2010), disaster management (Eckle-Elze and De Albuquerque, 2015) and disaster risk reduction (Scholz et al., 2018), earthquake damage assessment (Barrington et al., 2012; Kohns et al., 2021), deforestation detection (Arcanjo et al., 2016), archaeological prospection (Stewart et al., 2020), and many more. There are numerous applications that facilitate and sometimes (spatially and thematically) coor-

dinate contributions of VGI for specific use cases and goals. One example is the application MapSwipe (Herfort et al., 2017). With MapSwipe, volunteers can collaborate with humanitarian organizations to help map regions in the world where relevant data for preparedness, resilience and humanitarian response is missing. In the standard type of project, volunteers examine tiles of tessellated orthorectified aerial and satellite imagery from various sources in order to detect given features of interests (e.g. buildings) in them, thus helping to identify areas that need more detailed mapping (i.e. digitization of previously unmapped features).

Based on an adapted version of MapSwipe, we developed a web application to enable crowd-sourced mapping of ice-wedge polygons in aerial imagery. As part of the project, several different project designs were explored and put to test in order to find the best solution that balances the feasibility of the task by non-expert contributors with the usefulness and relevance of the mapping output for permafrost research. During preliminary tests, the most promising approach in terms of volunteer engagement and mapping efficiency proved to be a design in which volunteers were instructed to mark the approximate centroids of recognizable ice-wedge polygons. In this study, we demonstrate to what extent the output of such projects (i.e. the crowd-sourced centroids) can be used to derive accurate geomorphological and hydrological properties of the ice-wedge network.

## 60 2 Study regions

Two study regions were selected to test the application of VGI for the structural analysis of ice-wedge polygons. The two regions are very different in terms of climate, geomorphology, and soil characteristics (Fig.: A1). The first study region, Cape Blossom (CB), is located on the Baldwin Peninsula in western Alaska. This region is located near the transition zone between continuous and discontinuous permafrost (Jorgenson et al., 2008; Jongejans, 2017). It is characterized by ice-rich Pleistocene permafrost featuring massive ice wedges which form high- but in parts also low-center polygons at the surface (Strauss et al., 2017). The undulating landscape features a variety of thermoerosional valleys, lake basins and drained lake basins. Due to its proximity to this transition zone and relatively warm permafrost conditions, the region is particularly susceptible to climate warming (Strauss et al., 2017). The mean annual temperature in Kotzebue (approximately 20 km north of CB) is -5.05 °C, with annual precipitation of 280 mm (Alaska Climate Research Center, 2023). The ground at this region is dominated by marine, fluvial and glaciogenic fine-grained sediments (Hopkins et al., 1961) and is primarily vegetated by mosses and sedges.

The second study region, Blueberry Hills (BH), is located in the Northwest Territories of Canada within the Mackenzie Delta. It is located in the zone of continuous permafrost where the permafrost depth exceeds 700 m (Ehlers, 2011). This region is characterized by ice-rich permafrost and is undergoing major transformations due to climate warming (Van der Sluijs et al., 2018). Furthermore, the region is accommodating the largest concentration of infrastructure in the Canadian Arctic, which requires dedicated monitoring of permafrost changes (Van der Sluijs et al., 2018). The mean annual temperature at BH is around -8.6 °C, and it receives approximately 254 mm of precipitation annually (Climate Atlas of Canada, 2023). The vegetation at the region is dominated by mosses, sedges, and shrubs, including a notable presence of blueberries.

**Figure 1.** Study regions a) Overview map showing location of the study regions with permafrost zones (Obu et al., 2018) b) Cape Blossom study region with high resolution MACS imagery (Rettelbach et al., 2024) c) Blueberry Hills study region with UAV-aquired imagery (Mueller et al., 2024) d) Detail of Cape Blossom e) Detail of Blueberry Hills.

# 3 Material and methods

#### 3.1 Remote Sensing Data

The CB region was surveyed as part of an aircraft based campaign in 2021 that delivered high-resolution multi-spectral imagery of the permafrost landscapes using the Modular Aerial Camera System (MACS) (Grosse et al., 2021). The entire survey, conducted on June 25, 2021, covers approximately 25.22 km<sup>2</sup>. The data collection involved multiple flight lines, with the aircraft maintaining an altitude between approximately 1,490 and 1,510 m above ground level. The imagery captured by the MACS system was processed into four-band orthophotos (blue, green, red, and near-infrared) and DSMs both featuring a spatial resolution of 20 cm. Data post-processing was performed using photogrammetric software to produce high-resolution

orthomosaics (Rettelbach et al., 2024). In this study, a selected part from tile 11-2 of the CB sub-project 1 RGB orthophoto was used as the study region. The area of the study region corresponds to 0.714 km<sup>2</sup> (see Rettelbach et al., 2024).

The BH region was surveyed using the DJI Mini 2 drone as a Citizen Science activity with students from the Moose Kerr School in Aklavik. The survey was conducted on September 24, 2022, with the aim of creating a drone image based orthomosaic. The dataset consists of 3,557 individual Digital Negative (DNG) images with RGB color channels captured at 120 m flight height resulting in an average resolution of 4.97 cm/pixel. Later resampling in the course of photogrammetric post-processing resulted in a resolution of about 10 cm/pixel, covering an area of 1.58 km<sup>2</sup>. The Unoccupied Aerial Vehicle (UAV) system was equipped with a camera using a 12 megapixel CMOS sensor. To enhance the accuracy of the DSMs, we employed a novel spiral flight pattern developed within the UndercoverEisAgenten project and presented in Mueller et al. (2023). The images were post-processed using Agisoft Metashape 1.8.4, generating sparse and dense point clouds, DSMs, and orthomosaics, with altitude data adjusted using the local 2 m resolution ArcticDEM (Porter et al., 2022) for improved vertical accuracy. The dataset is openly available here (Mueller et al., 2024).

## 3.2 VGI data collection

The remote sensing data were analyzed by volunteers, primarily as part of workshops at schools and universities organized as mapping events ("mapathons"). The participants consisted of university level Geography students in Germany (two mapping events) and the Netherlands (one mapping event) and school students in grades 7–12 (approximate age range 12-18 years, seven mapping events) in German Higher Secondary Education (Gymnasium). Contributions to the mapping projects were made from a total number of 105 distinct user accounts.

All mapping events were preceded by an introduction adapted to the level of expected prior knowledge of the target group that comprised information on the topic of permafrost thawing, the formation of ice-wedge polygons, and the consequences of permafrost thawing for the environment and infrastructure. Subsequently, the students participated in mapping activities. The mapping projects differed between the mapping events in regard to area of interest (subsets of the two study regions) and task format. Following the micro-mapping approach, the larger areas of interest were split into smaller regularly shaped and equally sized micro-tasks. Participants mapped groups of spatially adjacent tasks one after another until either the area of interest was completely mapped or the mapping event ended.

While several task designs were tested with different audiences, this study is based on the output of point digitization tasks (see Appendix A). In the context of this task design, participants were instructed to mark the approximate centroids of ice-wedge polygons within the respective boundaries of each task area. Such digitization tasks aim to produce digital geographic objects (vector features) using given georeferenced image data (Albuquerque et al., 2016). In the crowd-sourced mapping application, the participants could interact with a web map by placing, modifying and deleting point markers within the task area boundaries.

Classification tasks of geographic crowd-sourcing – usually consisting in enriching georeferenced images with labels – are considered the task type with the lowest level of spatial cognitive complexity (Albuquerque et al., 2016). However, classifying ice-wedge polygons on aerial imagery of the Arctic surface has proven to be a considerably more challenging task for volunteer

contributors than e.g. the detection of buildings (Fritz et al., 2022). Our further experimentation demonstrated that point digitization tasks allowed volunteers to detect ice-wedge polygons with higher agreement (as an intrinsic indicator for accuracy) and at the same time in a more time-efficient manner. Beyond information on the presence of ice-wedge polygons (as the output of classification tasks), point digitization of polygon centers allows to quantify and locate individual polygon structures. While line digitization of ice-wedge polygons (i.e. the tracing of polygon outlines), on the other hand, would provide additional information on the polygon sizes and shapes, inital trials have shown that this task type overwhelms volunteer contributors and reduces the area to be mapped compared to point digitization. Assigning volunteer contributors with less difficult point digitization tasks while deriving further geomorphological and hydrological information through the network reconstruction method described in Sect. 3.3 was thus identified as a promising task design to monitoring Arctic permafrost with VGI.

For quality assurance of crowd-sourced data, it is common practice that multiple contributors map the same area. The individual contributions are then compared and aggregated to a collective result (Albuquerque et al., 2016). Here, each microtask was assigned to more than one contributor, each of them producing an individual point data set for the respective task area. These individual task results were merged into comprehensive vector point data sets covering the two study regions. To derive an "aggregated" result data set from the individual task results, it was necessary to cluster the point markings (see Sect. 3.3) to match the identified polygon centers before further analysis.

Along with the VGI data, an additional set of ice-wedge polygon centroids in the same two study regions was contributed by experts for the purpose of quality assessment. The study regions were analyzed by three authors of this paper analogously to the procedure of the VGI data, but in this case using the GIS software QGIS. It is referred to as the expert data set. For further validation of the Voronoi polygons in the reconstructed networks, a set of "ground truth" reference polygons were created through manual digitization by experts for subsets of the two study regions.

# 140 3.3 Data processing and analysis

135

To derive the target variables describing the geomorphological and hydrological characteristics of polygonal tundra, a specific multi-stage workflow was applied on both the volunteer-contributed and the expert-contributed ice-wedge polygon centroids (Fig.: 2). In contrast to the expert contributions, multiple volunteer contributions were collected for the same micro-tasks in order to ensure the quality of the results. It was thus necessary to aggregate the individual volunteer contributions of ice-wedge polygon centroids into average results, i.e. the mean locations of ice-wedge polygon centroids. To aggregate the individual results, clusters of the contributed ice-wedge polygon centroids were identified using DBSCAN (Density-Based Spatial Clustering of Applications with Noise), a density-based clustering algorithm originally developed by Ester et al. (1996) and the mean of each cluster was used for further analysis. One-point clusters were excluded, as the aim here was to ensure that an ice-wedge polygon was recognized by at least two volunteers. The centroids of the resulting clusters were considered as the crowd-contributed locations of ice-wedge polygon centroids.

**Figure 2.** Data processing and analysis workflow: Ice-wedge polygon networks are reconstructed from ice-wedge polygon centroids contributed by both volunteers and experts separately and validated against polygons digitized by experts. Geomorphological and hydrological parameters are derived from the reconstructed networks.

To establish the number of volunteer contributors needed to map the same region, we compared the positional accuracy of volunteer-contributed ice-wedge polygon centroids by cluster size (i.e. number of volunteer contributions establishing the centroid location) using a pairwise Tukey Honestly Significant Difference (HSD) test as implemented by the Python library statsmodel (Seabold and Perktold, 2010).

A second step of clustering needed to be applied on both the expert-contributed and the aggregated volunteer-contributed centroids to delineate ice-wedge polygon sub-networks, as it cannot be assumed that a study region is characterized by only a single connected ice-wedge polygon network. A combination of Delauny triangulation and alpha shape (Edelsbrunner et al., 1983) was used to identify the sub-networks in both study regions. An alpha value of 0.067 was chosen for this application. These steps were applied consistently across both research regions, and thus have the potential for replication in other contexts.

Voronoi diagrams were generated from both the expert-contributed and the aggregated crowd-sourced ice-wedge polygon centroids to reconstruct the polygonal ice-wedge networks. Cresto Aleina et al. (2013) and Ulrich et al. (2014) have demonstrated that automatically derived Thiessen polygons can represent ice-wedge networks with sufficient accuracy (i.e. a goodness of fit of  $R^2 = 0.84$  for the linear regression values of manually mapped polygon sizes against Thiessen polygon sizes according to Ulrich et al. (2014)). In our workflow we made use of the relationship between Delauny triangulation and Voronoi diagrams to derive polygon networks from the individual ice-wedge polygon centroids within each sub-network. Ice-wedge polygon

centroids at the edge of each (sub-) network were omitted as they would generate incorrect Thiessen polygons due to the lack of neighbors.

To assess the accuracy of the polygons generated from points digitized both by volunteers and experts through application of the Voronoi method, we compared them against a set of reference polygons whose outlines (not: centroids) were manually digitized by experts (see Appendix B1), considered as ground truth. These reference polygons were created by experts through visual interpretation of the same high-resolution aerial imagery, ensuring a high level of accuracy in delineating individual ice-wedge polygons. The validation process involved the calculation of four key metrics being precision, recall, F1-score, and Intersection over Union (IoU). Precision measures the proportion of correctly identified polygons among all polygons generated by the Voronoi method. Recall quantifies the proportion of correctly identified polygons out of all true polygons in the reference data. The F1-score provides a harmonic mean of precision and recall, offering a balanced measure of overall accuracy. IoU calculates the ratio of the intersection area to the union area between the predicted Voronoi polygon and its corresponding reference polygon reflecting the degree of overlap.

As the last step, geomorphological and hydrological parameters were derived from the network graphs of the two entire study regions and of subsets for the two regions used in the comparison with reference polygons. Geomorphological parameters include polygon area, perimeter, and distance to nearest neighbor. The latter describes the Euclidean distance from the center of each polygon to the centers of directly neighboring polygons per sub-network. The results were averaged for each polygon center. The hydrological properties were derived using the Python package NetworkX which has already been successfully used for the delineation of ice-wedge networks by Hagberg et al. (2008). In this study, NetworkX was used to calculate betweenness centrality, which quantifies the centrality of troughs within the ice-wedge network. The betweenness centrality is a measure of the importance of individual troughs for maintaining the flow of water within the network (Rettelbach et al., 2021). Thus, a high betweenness centrality indicate troughs that are likely to feature a increased water discharge and could be therefore be more susceptible to erosion.

#### 4 Results

# 4.1 VGI mapping results

For the CB data set, 21 volunteers mapped 10,337 points over five calendar days. 88 % of the points were mapped during two school visits (Fig.: 3). The remaining 12 % were mapped individually outside of any organized event. The aerial image illustrates that the ice-wedge polygons are evenly distributed over the study region, as are the contributed ice-wedge polygon centroids.

**Figure 3.** Volunteer-contributed points representing the approximate centroids of ice-wedge polygons for 1) Cape Blossom and 2) Blueberry Hills.

For the BH data set, 86 volunteers participated in the mapping process. A total number of 7,878 points (see Fig.: 3) were mapped over 29 calendar days. 76.61 % of the points were contributed during one of the mapping events at three different schools and one university. Most points were contributed by students from Higher Secondary Education (Gymnasium). The individual events resulted in different numbers of digitized points. The events were organized independent from each other at different locations. The exact format of each mapping event was slightly different, as was the number of participants. About one quarter of the points (23.39 %) were not mapped during any mapping event. In contrast to the CB study region, the aerial imagery clearly shows that ice-wedge polygons are distributed in spatial clusters over the BH study region. Accordingly, the contributed points form visually noticeable clusters within the study region. Furthermore, in both regions, the points contributed by multiple volunteers visibly form smaller clusters within the observable boundaries of ice-wedge polygons (see inset maps of Fig.: 3). This indicates that different volunteers contributing to the same mapping micro-tasks effectively identified the same ice-wedge polygons and approximately the same centroid locations.

#### 4.2 Ice-wedge polygon centroid quality assessment

One concern regarding VGI is the data quality and its fitness for a specific purpose (Mocnik et al., 2017). The purpose of generating the dataset of volunteer-contributed ice-wedge polygon centroids is to derive geomorphological and hydrological properties of the polygon networks in the two study regions. The dataset's fitness for purpose depends on (i) the feature completeness of the ice-wedge polygon centroids and (ii) their positional accuracy. Errors such as commission, omission, as well as misplacement of the centroids will influence the accuracy of the reconstructed networks and, consequently, of the derived properties.

In absence of ground truth data captured in situ, these two relevant dimensions of data quality are compared to a dataset of expert-contributed ice-wedge polygon centroids (see Sect. 3.2). While the expert-contributed data cannot be considered ground truth in the narrower sense, the extent of the deviation between volunteer- and expert-contributed data is expected to provide an indication of the data quality.

The feature completeness can be assessed by comparing the total number of ice-wedge polygon centroids in the volunteer and expert datasets. For the CB region, the number of clustered volunteer-contributed centroids (1,710) amounts to 88.74 % of the number of expert-contributed centroids (1,927). In the BH region, however, clustered volunteer-contributed centroids (769) only amount to 70.81 % of the number of expert-contributed data (1,086). The lower completeness of the volunteer-contributed dataset in BH, assessed against expert data, can be explained by the difference in the configuration of the polygon networks. Volunteers particularly often omitted ice-wedge polygons in areas at the borders of networks, and in smaller sub-networks (that characterize the BH region, see Section 4.3), whereas the single large network of CB as well as the larger sub-networks in BH are better represented.

Regarding the positional accuracy of the ice-wedge polygon centroid, the clustered volunteer-contributed centroids can be assessed by the deviation from the nearest expert-contributed centroid in space. For both study regions, the distributions of distances of all volunteer-contributed centroids from their nearest expert-contributed neighbor are clearly right-skewed, with rather small deviations for most of the centroids (Fig.: 4): The median distance is 1.29 m (mean: 1.56 m) in CB and 1.38 m

(mean: 2.66 m) in BH. For comparison: the approximate median distance between ice-wedge polygon centroids, based on the *aggregated* volunteer-contributed centroids, was determined as 19.34 m (CB) and 13.15 m (BH) respectively (see Table C1).

**Figure 4.** Distribution of distances between volunteer-contributed points and their nearest expert-contributed neighbors (as an indicator of positional accuracy) by study region.

With regard to the efficiency of the use of VGI in monitoring Arctic permafrost, it is vital to determine the optimal number of volunteers per micro-task required to map ice-wedge polygons with the sufficient quality. From the distributions of the distances between volunteer-contributed centroids and their nearest expert-contributed neighbor per cluster size (i.e., the number of volunteers that contributed to the centroid), it can be seen that, as a general trend, the higher the number of contributors per polygon, the better the positional accuracy (Fig.: 5). For CB, the positional accuracy of the volunteer-contributed centroids does not improve significantly after surpassing the number of five contributions per polygon (see Table C2). For the BH region, the data does not support clear conclusions due to the relatively low number of volunteer-contributed points of medium cluster sizes (see Table C3).

#### Occurrences of Each Cluster Size

|   |                 | 2   | 3   | 4   | 5   | 6   | 7   | 8   | 9+  |
|---|-----------------|-----|-----|-----|-----|-----|-----|-----|-----|
|   | Cape Blossom    | 90  | 106 | 210 | 263 | 282 | 218 | 244 | 137 |
| 1 | Blueberry Hills | 170 | 102 | 51  | 50  | 18  | 15  | 17  | 272 |

**Figure 5.** Distributions of distances between volunteer-contributed points and their nearest expert-contributed neighbors (as an indicator of positional accuracy) by cluster size. Cluster size refers to the number of volunteers that mapped a specific polygon.

#### 4.3 Network reconstruction

Following the generic workflow described in Sect. 3.3, ice-wedge polygon networks could be effectively reconstructed from volunteer-contributed ice-wedge polygon centroids in both of the study regions, despite their very different characteristics. For both of the study regions, the resulting networks appear plausible upon visual inspection. In the CB region with its rather evenly distributed ice-wedge polygons visible on the surface across the entire study region, the proposed approach has a single contiguous network of 1,490 polygons as an output (see Fig.: 6).

**Figure 6.** Voronoi diagrams derived from clustered volunteer-contributed ice-wedge polygon centroids, representing the reconstructed ice-wedge network for 1) Cape Blossom and 2) Blueberry Hills. In the BH study region, smaller sub-networks are entirely omitted where polygons could not be formed due to missing neighbors.

In the BH region, the reconstructed network is in agreement with the clearly clustered occurrence of ice-wedge polygons in specific regions of the study region. The resulting network consists of 21 different sub-networks with an average number of 16 polygons, ranging from one single polygon to 133 per sub-network (see Fig.: 6).

Comparing the originally contributed points (see Fig.: 3) with the resulting network (see Fig.: 6), it becomes, however, evident that due to the necessary removal of centroids at the edge of networks (see Sect. 3.3), some smaller sub-networks may be entirely omitted.

## 4.4 Voronoi Network Validation

250

265

The results of the accuracy assessment of reconstructed ice-wedge polygons derived both from centroids contributed by volunteers and by experts against polygons digitized by experts (see Sec. 3.3) indicate a reasonably good agreement between the Voronoi polygons and the reference data across the two study regions and data sources (Tab.: 1).

**Table 1.** Accuracy assessment of the Voronoi polygons generated from volunteer- and expert-derived polygon centers at the Blueberry Hills and Cape Blossom study regions. Metrics include Precision, Recall, F1-Score, and Median Intersection-over-Union (IoU).

| Metric     | Cape Blossom |        |           | Blueberry Hills |  |  |
|------------|--------------|--------|-----------|-----------------|--|--|
|            | volunteer    | expert | volunteer | expert          |  |  |
| Precision  | 0.77         | 0.77   | 0.65      | 0.72            |  |  |
| Recall     | 0.81         | 0.88   | 0.73      | 0.82            |  |  |
| F1-Score   | 0.79         | 0.83   | 0.69      | 0.77            |  |  |
| Median IoU | 0.71         | 0.72   | 0.57      | 0.67            |  |  |

In general, the expert-derived polygons exhibit higher values across the accuracy metrics compared to the VGI-derived polygons. In the CB region, a difference in the F1-score is driven by a better recall in the expert set (experts: 0.88, VGI: 0.81), whereas precision and median IoU do not vary substantially between the comparison data sets. This concludes that volunteer mappers failed to detect more ice-wedge polygon centroids than experts, but did not commit more errors in the points that they did identify than the experts. However, the difference in the values is more pronounced, and extends to the precision (experts: 0.72, VGI: 0.65). Despite the overall good performance, the moderate IoU values (CB: 0.71, BH: 0.57) suggest some discrepancies in the shapes and sizes of the polygons. Visual inspection revealed that these discrepancies are more pronounced in regions with heterogeneous landscapes, where the underlying environmental factors influencing ice-wedge polygon morphology are more complex.

#### 4.5 Geomorphological properties

The workflow described in Sect. 3.3 results in CB being represented as a single network of 1,490 ice-wedge polygons in the volunteer dataset and 1,679 in the expert dataset (Table C1). BH consists of multiple sub-networks (21 in the volunteer dataset

and 25 in the dataset mapped by experts), encompassing 1–133 and 1–140 polygons respectively. Both the average number of polygons per sub-network and the number of sub-networks in this study region are lower in the volunteer dataset than in the expert dataset. On the average, ice-wedge polygons in BH are smaller than the ones in CB (Fig.: 7). The median polygon area is 304 m<sup>2</sup> for CB and 147 m<sup>2</sup> for BH, with the expert-contributed data set showing a slightly higher median area for BH (160 m<sup>2</sup>) and a similar one for CB (303 m<sup>2</sup>). The ice-wedge polygons have a median perimeter of 68 m in CB and 48 m in BH, differing by only 1–2 m from the values in the expert data set. The median distance between neighboring centroids is approximately 19 m for CB and 13 m for BH, which is consistent with the expert dataset. The relative standard deviation values for all parameters are in high agreement between the expert- and volunteer-contributed datasets.

**Figure 7.** Distribution of values for a) polygon area (in m<sup>2</sup>), b) perimeter (in m) and c) the distance to the neighboring ice-wedge polygon centroids (in m). The plot compares the study regions and the volunteer- (red) and expert-contributed (grey) dataset each.

The same statistics computed for subsets of each study region allow for evaluation against reference polygons (described in Sect. 3.3). The relevant subsets covered by the reference polygon data are shown in Appendix B and include about 100 ice-wedge polygons in CB and 150 in BH. The polygon area exhibits the most discrepancies between the three datasets (Table 2). Both the volunteer- and expert-contributed datasets overestimate the polygon area by an average of 10 to 30 m<sup>2</sup> while the data set contributed by volunteers is somewhat closer to the reference data set for both study regions. The polygon perimeter is accurately represented in both the expert- and volunteer-contributed datasets for CB with only minor deviations from the reference dataset (mean deviations: 0–2 m). In BH, the perimeters are overestimated by about 3–4 m on average by both volunteers and experts. In both study regions, only minor deviations are observed between the reference, volunteer and expert polygons as concerns the distance between neighboring centroids.

**Table 2.** Comparison of Ice-Wedge Polygon Statistics between ice-wedge polygons derived from centroids digitized by volunteers and experts, and polygons from outlines manually digitized by experts (reference) for two Arctic study regions (Cape Blossom and Blueberry Hills). The metrics include polygon count, area and perimeter measurements, and distances between neighboring ice-wedge polygon centroids and are calculated for a subset area specified by the extent of the mapped reference polygons.

|                                  | Cape Blossom |         |           | Blueberry Hills |         |           |  |
|----------------------------------|--------------|---------|-----------|-----------------|---------|-----------|--|
|                                  | volunteer    | expert  | reference | volunteer       | expert  | reference |  |
| No. of polygons                  | 108          | 100     | 100       | 138             | 150     | 159       |  |
| Polygon area (m <sup>2</sup> )   |              |         |           |                 |         |           |  |
| mean                             | 301.54       | 321.31  | 289.65    | 162.68          | 164.33  | 140.31    |  |
| median                           | 292.99       | 319.33  | 270.59    | 153.85          | 155.17  | 126.25    |  |
| standard deviation (std)         | 82.38        | 100.82  | 114.93    | 66.13           | 59.02   | 90.61     |  |
| relative std                     | 27.32 %      | 31.38 % | 39.68 %   | 40.65 %         | 35.91 % | 64.58 %   |  |
| Polygon perimeter (m)            |              |         |           |                 |         |           |  |
| mean                             | 67.29        | 69.37   | 67.20     | 50.30           | 49.79   | 45.26     |  |
| median                           | 66.62        | 69.83   | 66.53     | 49.30           | 49.27   | 44.44     |  |
| std                              | 8.36         | 10.44   | 13.23     | 9.64            | 8.77    | 14.11     |  |
| relative std                     | 12.43 %      | 15.05 % | 19.69 %   | 19.17 %         | 17.61 % | 31.17 %   |  |
| Distance between neighboring cer | ntroids (m)  |         |           |                 |         |           |  |
| mean                             | 19.26        | 19.78   | 19.16     | 14.15           | 13.90   | 13.26     |  |
| median                           | 18.90        | 19.92   | 19.23     | 14.21           | 13.81   | 13.39     |  |
| std                              | 2.38         | 2.77    | 2.74      | 2.27            | 2.26    | 2.81      |  |
| relative std                     | 12.33 %      | 13.99 % | 14.29 %   | 16.07 %         | 16.27 % | 21.24 %   |  |

# 4.6 Hydrological properties

Betweenness centrality provides a measure of the importance of individual channels for water drainage within hydrological networks (Marra et al., 2014). Channels with high centrality act as critical connectors, linking otherwise isolated parts of the network and thereby playing a key role in maintaining or enabling overall drainage. In the context of the hydrological function of ice-wedge polygon networks, through segments with high centrality are likely to carry disproportionately large water fluxes, as they concentrate flow. Consequently, they play an important role in the transport of dissolved nutrients and other substances, while also being more susceptible to enhanced erosion and thermokarst development (Rettelbach et al., 2021).

In CB, edges with high betweenness centrality values derived from the volunteer-contributed graphs, i.e., troughs potentially more affected by erosion, are located at the center of the network (Fig.: 8). The polygons delimited by edges with high betweenness centrality values often have relatively large areas and perimeters. When visually comparing these troughs with remote sensing data, they often coincide with areas of surface water occurrence (dark polygon centers).

**Figure 8.** Visualization of the betweenness centrality values of the edges in the ice-wedge networks reconstructed from volunteer-contributed ice-wedge polygon centroids for 1) Cape Blossom and 2) Blueberry Hills. Especially the inset of CB shows visible drainage paths underneath edges of relatively high betweenness centrality.

In BH, the maximum betweenness centrality value is approximately ten times lower than in CB due to the smaller size of the sub-networks (Fig.: 8). For the same reason, the maximum betweenness centrality values of edges differ within the same region between the sub-networks, and are generally higher in larger sub-networks. Similar to CB, edges of high betweenness centrality are located in the center of the sub-networks. In addition, visual inspection intriguingly shows visible drainage path underneath edges of relatively high betweenness centrality.

#### 5 Discussion

# 5.1 Application

Mapping the polygon network provides a primary understanding of polygonal terrain and the results can be used for a variety of applications, such as determining where ice wedges tend to form by comparing their distribution with landscape parameters like slope and surficial deposits (Frappier and Lacelle, 2021). Furthermore, it enables quantitative characterization of ice-wedge polygons conditions and spatial properties, which provides critical insights into past and current landscape shaping and altering processes. For instance, measuring the angles and regularity of the network based on spatial patterns of polygon intersections, helps determine the maturity of ice-wedge networks and how they evolve in different geomorphological settings (Haltigin et al., 2012; Sletten et al., 2003; Frappier and Lacelle, 2021). Moreover, 3D subsurface models derived from the mapped polygon networks allow for estimating wedge ice volume, a key factor in predicting thermokarst formation as permafrost degrades (Ulrich et al., 2014; Couture and Pollard, 2017).

Additionally, once the polygon network has been delineated, it can be used to effectively extract different properties of the polygonal terrain, contributing to the understanding of surface and subsurface processes occurring at the local scale (i.e., intra- or inter-polygons). The microtopography of the polygons can be extracted from high resolution digital elevation models (DEMs) (Abolt et al., 2019; Abolt and Young, 2019) or ground displacement can be measured from InSAR data (Short and Fraser, 2023). These data can inform on the state of the ice wedges, as degradation typically leads to subsidence of the soil above the ice wedge, progressively forming high-centered polygons (Kanevskiy et al., 2017; Jorgenson et al., 2015). Similarly, vegetation and wetness indices can be extracted from spectral imagery to understand surface and subsurface wetness as well as vegetation distribution patterns, which control a broad range of interactions between the ground and the atmosphere (e.g., Zhang et al. (2018); Morse and Burn (2013); Langer et al. (2011a, b)).

The ability to derive these spatial metrics and characteristics not only improves our understanding of ice-wedge distribution and condition, but also supports broader applications such as extrapolating these findings to other regions and contributing to predictive models of ice-wedge polygon evolution (Jorgenson et al., 2015; Zhang et al., 2018; Liljedahl et al., 2016; O'Neill et al., 2019).

#### 5.2 Limitations and potentials

Reconstructing the ice-wedge polygon network from volunteer-generated ice-wedge polygon centroids can be a viable alternative to automated workflows, particularly if otherwise necessary data (such as a high resolution digital elevation model) is not available. The high similarity between the reconstructed networks and the polygon networks observed in RGB aerial images, despite significant differences in the ice-wedge polygon network configurations, demonstrates the effectiveness of the presented approach. The combination of low-effort, time-efficient crowd-sourced mapping by point digitizing with the reconstruction of networks can be a suitable solution to derive geomorphological and hydrological properties of polygonal tundra under different landscape conditions. However, the spatial coverage that can be reached by such an approach is limited by i) the number of participating volunteers, ii) the number of tasks that each volunteer is willing to complete, iii) the overall person-hours of volunteer contributions that can be mobilized for the the crowd-sourced mapping process, and iv) the time needed for the completion of each task. In the case of the volunteer-contributed data used in this study, the participants needed a median time of 2.9 seconds for each digitized point. Observations in early trials indicated that point digitization was the most time-efficient one of the tested task designs. In a trial session to evaluate different task types, participants needed a median of 2 seconds per 1,000 m<sup>2</sup> of area (compared to 2.8 seconds in a classification task design). It is important to note that the mapping speed may vary strongly with factors such as the visibility of the ice-wedge polygons in the imagery and the mapping experience of the contributors. To make efficient use of volunteer contributions, it is important to optimize the number of participants that each task is assigned to. In Sect. 4.2, we identify a number of five contributions per task as sufficient for assuring a high quality of the aggregated crowd-sourced data. This number is in line with previous findings on the optimal number of volunteer contributors to tasks of building classification from aerial imagery (Herfort, 2018).

In relation to the efficiency of the mapping method presented, the additional effort required to recruit volunteers for a crowd-sourcing activity must also be taken into account. Volunteer engagement clearly demands communication, outreach, provision of context information and motivation. In our case, this entailed among other outreach to schools and teachers, creation of teaching material, and the preparation of mapping events, with 13 person months allocated to community involvement and training in the framework of a larger citizen science project. As Huang et al. (2023) highlight, recruiting volunteers for crowd-sourced mapping tasks can be particularly challenging when the features of interest are highly specific and unfamiliar to most people, such as ice-wedge polygons. Sustaining volunteer contributions beyond individual mapping events across larger spatial extents is therefore likely to require tailored engagement strategies. Potential approaches include: (a) substantially broadening outreach efforts to educational institutions, while equipping teachers with ready-to-use teaching materials to facilitate in-class mapping sessions; (b) integrating ice-wedge polygon mapping projects into established crowd-sourced mapping platforms with active communities and providing interactive tutorials to support self-guided learning; and (c) incorporating gamification elements such as leaderboards, badges, and activity streaks to foster motivation and sustained participation.

The application for crowd-sourced mapping of ice-wedge polygons was designed to enable contributions by people without particular domain expertise, with some information and context provided through teaching materials. While prior knowledge allowed for providing more in-depth contextualization to geography students at the respective events, the concrete mapping

task was designed to depend on general pattern recognition skills. The majority of results were generated by secondary school students. This study does not provide a comparison of quality of results for different user groups from a controlled experimental setting. Such a comparison might be useful to inform any further extension of the volunteer base.

For (volunteer) contributors to be able to accurately identify polygonal structures, the quality of imagery used as a base in crowd-sourced mapping is critical. Factors such as lighting and atmospheric conditions during image acquisition significantly influence the clarity and contrast needed for recognition (O'Connor et al., 2017). In the case of this study, some areas in the BH dataset contain small blurry sections due to the UAV flight geometry and lighting conditions. However, it is well-documented that human interpreters can often discern and infer subtle structures within images even if the image quality is strongly reduced (Wang et al., 2024). This ability is providing a distinct advantage over traditional automated methods, which are more dependent on high contrast and clear image conditions for effective structure recognition. With the advancement of modern machine learning techniques in image analysis, such as Convolutional Neural Networks (CNNs) the differences between human and automated structure recognition abilities are increasingly fading (e.g. Wei et al., 2024). There is an interest in assessing how these approaches compare to human interpretation in accuracy and reliability (e.g. Lake et al., 2015). The VGI method presented here could contribute valuable insights, offering a baseline dataset that could serve both training and validation, potentially further enhancing machine learning models.

With respect to spatial resolution, this study does not experimentally compare mapping outcomes generated from 20 cm (CB) and 10 cm (BH) resolution imagery. It is to be assumed that resolution – at this level – is not a decisive factor for the results. The BH orthomosaic was indeed downsampled from 5 cm to 10 cm for practical reasons, as the native resolution was deemed excessive for the given mapping task on visual inspection. Nonetheless, access to sufficiently high-resolution imagery remains important: given the typical dimensions of ice-wedge polygons and the width of their delimiting troughs, crowd-sourced mapping of these features would not be feasible with coarser-resolution data.

The assessment in Sect. 4.2 demonstrates that the quality, and particularly the completeness of the crowd-sourced ice-wedge polygon centroids depends on the configuration of the network, with more centroids omitted in areas of borders of networks and in smaller sub-networks. This may be explained by the fact that the identification of smaller clusters of ice-wedge polygons within regions with only few features of interest requires a more systematic approach inspecting the imagery of a specific micro-task. In addition, our approach required to omit edge polygons from sub-networks (see Sect. 4.3). For smaller networks, edge removal will result in a larger proportion of the polygons identified by volunteers being removed from the output data, and may even lead to entire sub-networks being omitted. Therefore, results may be more accurate for regions with large contiguous areas of ice-wedge polygons (such as CB) than for regions with dispersed clusters of smaller ice-wedge networks.

The application of the Voronoi method to reconstruct polygon networks from approximate centroids means that the resulting ice-wedge polygon boundaries are inclusive of the troughs. This may partially explain differences both in average area and perimeter, when the reconstructed crowd-sourced polygon boundaries are compared to reference polygon boundaries that were captured exclusive of troughs, as shown in Sect. 4.5. In general, our proposed method does not allow for gaining information about trough sizes.

It has been demonstrated that natural crack mosaics in drying clay represent a random tessellation that evolves with repeated wetting and drying cycles towards a Voronoi mosaic that minimizes internal energy (Haque et al., 2023). Although there are similarities between the repetitive cracking processes in drying and revetted clays and the formation of thermal contraction cracks in frozen soils, it is not clear whether this analogy is generally transferable to formation of Voronoi structures. So far reconstructing ice-wedge networks using Voronoi diagrams are reported to be particularly effective for orthogonal polygons featuring mainly rectangular or hexagonal shapes. These structures are often associated with relatively early stages of ice-wedge polygon development (low- and flat-centered polygons). It has been demonstrated that these types of polygons can be well-represented by Voronoi tessellations (Cresto Aleina et al., 2013; Ulrich et al., 2014).

However, as the ice-wedge network evolves, forming secondary and higher order cracks, the ice-wedge network can become more irregular. This could limit the ability of Voronoi diagrams to represent all components of mature polygon networks. In consequence this would result in overestimation of polygon areas and underestimation of wedge-ice volume (Bernard-Grand'Maison and Pollard, 2018). This leads to the need to better understand the structural formation of ice-wedge networks. The presented VGI method could be used to better understand to what extent the real polygon network differs from the one reconstructed by the Voronoi diagram. This could be added to the VGI concept, which could be extended by a function to directly map trough lines in addition to ice-wedge polygon centroids. The degree of deviation could deliver important insights into the evolution stage of of the ice-wedge network. However, extending the VGI concept with polygon or line digitization tasks would require further research into the consequences of such extension on mapping efficiency, accuracy, and volunteer engagement.

While our proposed method for network reconstruction from crowd-sourced ice-wedge polygon centroids is specifically tailored to the monitoring of ice-wedge polygons, the underlying approach of micro-task-based crowd-sourced mapping holds potential for a wider range of environmental monitoring applications. In particular, it could be applied to other permafrost landforms such as pingos or thaw slumps, provided these features are sufficiently visible in available imagery to non-expert volunteers. A key limitation, however, lies in the spatial distribution of the target features: if they occur too sparsely within the designated mapping area, sustaining volunteer engagement may become increasingly difficult.

#### 415 6 Conclusions

395

The proposed methodology enables the use of VGI for monitoring Arctic permafrost. Reconstruction of ice-wedge networks from crowd-sourced ice-wedge polygon centroids can be a viable alternative to automated workflows for deriving geomorphological and hydrological properties of permafrost landscapes, especially when elevation data of the necessary horizontal resolution and vertical accuracy is unavailable.

Volunteers are able to efficiently map ice-wedge polygon centroids with reasonable accuracy. Volunteer-contributed point data in both study areas show acceptable variation in completeness (CB: 88.74 %, BH: 70.81 %) and positional accuracy (median distance to nearest expert-mapped centroid – CB: 1.29 m, BH: 1.56 m) when compared to expert-contributed points. The quality of the volunteer-contributed data, however, depends on the number of volunteers contributing to each mapping task

and on the configuration of the ice-wedge network (e.g. the spatial distribution of ice-wedge polygons) in the study region.

Based on our findings, we recommend to assign five volunteers to each micro-task to achieve high quality results. The VGI data match those generated by experts best in evenly distributed networks like those at Cape Blossom.

The point data have been successfully utilized to reconstruct ice-wedge polygon networks. A visual inspection of the derived network shows a high level of agreement with the actual network structure, consistent with previous studies by Cresto Aleina et al. (2013) and Ulrich et al. (2014), which also employ Voronoi diagrams to reconstruct ice-wedge networks. Compared with reference polygons traced by experts, the reconstructed networks reach high values of accuracy, particularly in evenly distributed networks (F1-Score: 0.79, Median IoU: 0.71), and with only a small difference between networks reconstructed from volunteer- and from expert-contributed centroids. Our findings demonstrate that using the Voronoi characteristics of ice-wedge polygons can effectively simplify the mapping process, enabling volunteers to complete the task with high quality and comparatively low effort.

Quantitative statistical descriptors on the geomorphology (polygon area, perimeter, and distance between neighboring centroids) and hydrology (betweenness centrality) were successfully derived from the reconstructed networks created from volunteer-contributed ice-wedge polygon centroids. These statistics allow comparing landscape differences spatially and monitoring changes over time. Moreover, the data generated can be utilized in land surface modeling schemes that incorporate information of aggregated landscape units to simulate sub-grid scale thermal and hydrological processes.

Code and data availability. A high-resolution UAV orthomosaic that served as a basis for crowd-sourced permafrost mapping of the Blueberry Hills study region is published as Mueller et al. (2024), the aerial MACS dataset for the Cape Blossom study region is made available as Rettelbach et al. (2024). Datasets of crowd-sourced and expert generated ice-wedge polygon centroids are available with Walz et al. (2025). Python scripts for data processing and analysis as described in Sect. 3.3 are available with Walz (2025). The availability of the web application used for crowd-sourced mapping is described in Appendix A.

# 445 Appendix A: Crowd-sourced mapping application

The VGI used in this study, that is, the approximate locations of ice-wedge polygon centers sourced by the crowd, was collected by the participants in the UndercoverEisAgenten project with the help of a web application <sup>1</sup> that was developed as part of the project.

<sup>1</sup> https://crowdmap.undercovereisagenten.org/

Figure A1. Screenshot of an ice-wedge polygon centroid digitization project in the crowd-sourced mapping application

This web application is based in large part on MapSwipe <sup>2</sup>, an existing application for crowd-sourced mapping in humanitarian use cases. While MapSwipe was originally developed as a mobile application for iOS and Android, the UndercoverEis-Agenten app uses a web based user interface made with the JavaScript framework Vue. The newly developed web application serves as a client for an adapted version of the MapSwipe back-end (see A2). One of the major advancements is that the UndercoverEisAgenten application allows for point digitization mapping tasks both in the user interface as well as in back-end workflows to create mapping projects and process mapping results. The web client directly communicates only with a Firebase Realtime database to load crowd-sourced mapping projects and tasks, and to save the results, i.e. the volunteer contributions. Using Firebase Realtime database ensures horizontal scaling, so that there is practically no limit of concurrent users of the app. Tasks are assigned to the user by random selection among the groups of tasks with the highest number of required contributions remaining. The mapping results are regularly transferred from the Firebase Realtime database to a Postgres database

<sup>&</sup>lt;sup>2</sup>https://mapswipe.org/

by Python workers for more sustainable and efficient storage and processing of large amounts of data. Mapping projects are drafted with a Manager Dashboard. Python workers regularly create projects from drafts and generate statistics from the results of ongoing projects stored in the Postgres database. Results and statistics are provided via an API in appropriate open formats (comma-separated value and GeoJSON files). Based on the UndercoverEisAgenten web client, MapSwipe was recently expanded to include a web interface as well. The MapSwipe web client source code is licensed under GPL-3.0 and is available at https://github.com/mapswipe/mapswipe-web. The back-end is available at https://github.com/mapswipe/python-mapswipe-workers under Apache-2.0 license.

**Figure A2.** MapSwipe architecture diagram showing the interaction between the mapping clients, the Firebase Realtime database and the backend.

# Appendix B: Visual Comparison and Validation of Voronoi Polygons and Reference Data

Figure B1 provides a visual comparison of the generated Voronoi polygons against the manually digitized reference polygons for both study regions (Cape Blossom and Blueberry Hills) and data sources (VGI and expert).

- Subfigure (a) illustrates the Voronoi polygons derived from VGI (pink) overlaid on the reference polygons (light blue) for the Cape Blossom region, with a median IoU of 0.71. The visual alignment between the Voronoi and reference polygons appears to be better in this case compared to Blueberry Hills, likely due to the more homogeneous landscape characteristics of the Cape Blossom region.
- Subfigure (b) shows the expert-derived Voronoi polygons for Cape Blossom (yellow), which achieve the highest median IoU of 0.72. The visual comparison confirms the high accuracy, with the Voronoi polygons closely matching the reference polygons across most of the region.

- Subfigure (c) depicts the Voronoi polygons derived from VGI (pink) overlaid on the reference polygons (light blue) for the Blueberry Hills region. The accompanying histogram shows the distribution of Intersection over Union (IoU) scores, with a median IoU of 0.57. Visually, the Voronoi polygons generally align with the reference polygons, but some discrepancies in shape and size are evident, particularly in areas with more complex terrain features.
- Subfigure (d) presents the expert-derived Voronoi polygons (yellow) for the same region. The median IoU in this case is 0.67, indicating a better agreement with the reference data compared to the VGI-derived polygons. The visual comparison supports this observation, showing a closer correspondence between the predicted and reference polygons.

**Figure B1.** Visual comparison of Voronoi polygons and reference data. (a) VGI-derived polygons at Cape Blossom. (b) Expert-derived polygons at Cape Blossom. (c) VGI-derived polygons at Blueberry Hills. (d) Expert-derived polygons at Blueberry Hills.

The figure highlights the influence of landscape heterogeneity and data source accuracy on the performance of the Voronoi method. Areas with complex terrain and VGI-derived polygon centers tend to exhibit lower IoU values and more visual discrepancies compared to areas with homogeneous landscapes and expert-derived centers. This visual analysis complements the

quantitative assessment presented in Table 1, providing a more comprehensive understanding of the accuracy and limitations of the Voronoi approach for mapping ice-wedge polygons.

# **Appendix C: Tables**

Table C1. Comparison of Polygon Statistics for Different Study regions

|                                 | Cape Blo              | ossom              | Blueberry Hills       |                    |  |  |
|---------------------------------|-----------------------|--------------------|-----------------------|--------------------|--|--|
|                                 | volunteer-contributed | expert-contributed | volunteer-contributed | expert-contributed |  |  |
| No. of sub-networks             | 1                     | 1                  | 21                    | 25                 |  |  |
| No. of polygons per sub-network | k                     |                    |                       |                    |  |  |
| minimum                         | 1490                  | 1679               | 1                     | 1                  |  |  |
| maximum                         | 1490                  | 1679               | 133                   | 140                |  |  |
| mean                            | 1490                  | 1679               | 16.85                 | 19.68              |  |  |
| Polygon area (m <sup>2</sup> )  |                       |                    |                       |                    |  |  |
| mean                            | 322.20                | 315.94             | 156.09                | 174.46             |  |  |
| median                          | 303.85                | 303.44             | 146.95                | 159.84             |  |  |
| standard deviation (std)        | 113.79                | 109.46             | 64.68                 | 71.76              |  |  |
| relative std                    | 35.32 %               | 34.65 %            | 41.44 %               | 41.13 %            |  |  |
| Polygon perimeter (m)           |                       |                    |                       |                    |  |  |
| mean                            | 69.28                 | 68.36              | 49.07                 | 51.38              |  |  |
| median                          | 68.34                 | 67.99              | 48.48                 | 50.15              |  |  |
| std                             | 11.11                 | 10.88              | 9.74                  | 10.01              |  |  |
| relative std                    | 16.04 %               | 15.92 %            | 19.85 %               | 19.49 %            |  |  |
| Distance between neighboring c  | entroids (m)          |                    |                       |                    |  |  |
| mean                            | 19.57                 | 19.34              | 13.32                 | 14.08              |  |  |
| median                          | 19.34                 | 19.20              | 13.15                 | 13.70              |  |  |
| std                             | 3.02                  | 3.01               | 2.53                  | 2.74               |  |  |
| relative std                    | 15.44 %               | 15.59 %            | 19.03 %               | 19.49 %            |  |  |

**Table C2.** Cape Blossom: Statistical Results of Post-hoc Test (Tukey HSD) for differences in medium distance between volunteer-contributed points and their nearest expert-contributed neighbors (as indicator of positional accuracy) between groups of different cluster sizes

| Cluster size 1 | Cluster size 2 | Mean Diff | p-adj  | Lower   | Upper   | Reject |
|----------------|----------------|-----------|--------|---------|---------|--------|
| 2              | 3              | -0.4526   | 0.0394 | -0.8936 | -0.0116 | True   |
| 2              | 4              | -0.6758   | 0.0000 | -1.0634 | -0.2882 | True   |
| 2              | 5              | -0.8182   | 0.0000 | -1.1940 | -0.4425 | True   |
| 2              | 6              | -0.9245   | 0.0000 | -1.2969 | -0.5520 | True   |
| 2              | 7              | -1.0173   | 0.0000 | -1.4028 | -0.6318 | True   |
| 2              | 8              | -1.0242   | 0.0000 | -1.4036 | -0.6447 | True   |
| 2              | 9+             | -0.8095   | 0.0000 | -1.2270 | -0.3921 | True   |
| 3              | 4              | -0.2232   | 0.5872 | -0.5898 | 0.1434  | False  |
| 3              | 5              | -0.3656   | 0.0371 | -0.7196 | -0.0117 | True   |
| 3              | 6              | -0.4718   | 0.0012 | -0.8224 | -0.1213 | True   |
| 3              | 7              | -0.5647   | 0.0001 | -0.9290 | -0.2003 | True   |
| 3              | 8              | -0.5715   | 0.0000 | -0.9294 | -0.2136 | True   |
| 3              | 9+             | -0.3569   | 0.1167 | -0.7549 | 0.0410  | False  |
| 4              | 5              | -0.1424   | 0.7977 | -0.4272 | 0.1423  | False  |
| 4              | 6              | -0.2487   | 0.1259 | -0.5291 | 0.0318  | False  |
| 4              | 7              | -0.3415   | 0.0119 | -0.6390 | -0.0440 | True   |
| 4              | 8              | -0.3484   | 0.0066 | -0.6380 | -0.0588 | True   |
| 4              | 9+             | -0.1338   | 0.9317 | -0.4716 | 0.2041  | False  |
| 5              | 6              | -0.1062   | 0.9254 | -0.3700 | 0.1575  | False  |
| 5              | 7              | -0.1990   | 0.3872 | -0.4808 | 0.0828  | False  |
| 5              | 8              | -0.2059   | 0.3023 | -0.4794 | 0.0675  | False  |
| 5              | 9+             | 0.0087    | 1.0000 | -0.3155 | 0.3329  | False  |
| 6              | 7              | -0.0928   | 0.9722 | -0.3703 | 0.1847  | False  |
| 6              | 8              | -0.0997   | 0.9515 | -0.3687 | 0.1693  | False  |
| 6              | 9+             | 0.1149    | 0.9593 | -0.2055 | 0.4353  | False  |
| 7              | 8              | -0.0069   | 1.0000 | -0.2936 | 0.2798  | False  |
| 7              | 9+             | 0.2077    | 0.5653 | -0.1277 | 0.5432  | False  |
| 8              | 9+             | 0.2146    | 0.4938 | -0.1139 | 0.5431  | False  |

**Table C3.** Blueberry Hills: Statistical Results of Post-hoc Test (Tukey HSD) for differences in medium distance to between volunteer-contributed points and their nearest expert-contributed neighbors (as indicator of positional accuracy) between groups of different cluster sizes

|   | Cluster size 2 | Mean Diff | p-adj  | Lower   | Upper   | Reject |
|---|----------------|-----------|--------|---------|---------|--------|
| 2 | 3              | -0.8823   | 0.1319 | -1.8848 | 0.1203  | False  |
| 2 | 4              | -1.5442   | 0.0063 | -2.8222 | -0.2662 | True   |
| 2 | 5              | -2.0151   | 0.0001 | -3.3029 | -0.7273 | True   |
| 2 | 6              | -2.7012   | 0.0010 | -4.6853 | -0.7171 | True   |
| 2 | 7              | -1.3433   | 0.5554 | -3.4993 | 0.8128  | False  |
| 2 | 8              | -1.3566   | 0.4653 | -3.3927 | 0.6796  | False  |
| 2 | 9+             | -2.5678   | 0.0000 | -3.3504 | -1.7852 | True   |
| 3 | 4              | -0.6620   | 0.8253 | -2.0348 | 0.7108  | False  |
| 3 | 5              | -1.1328   | 0.2004 | -2.5147 | 0.2491  | False  |
| 3 | 6              | -1.8189   | 0.1236 | -3.8654 | 0.2275  | False  |
| 3 | 7              | -0.4610   | 0.9984 | -2.6746 | 1.7526  | False  |
| 3 | 8              | -0.4743   | 0.9973 | -2.5713 | 1.6227  | False  |
| 3 | 9+             | -1.6855   | 0.0000 | -2.6149 | -0.7562 | True   |
| 4 | 5              | -0.4709   | 0.9862 | -2.0639 | 1.1222  | False  |
| 4 | 6              | -1.1570   | 0.7486 | -3.3516 | 1.0376  | False  |
| 4 | 7              | 0.2010    | 1.0000 | -2.1502 | 2.5522  | False  |
| 4 | 8              | 0.1877    | 1.0000 | -2.0541 | 2.4294  | False  |
| 4 | 9+             | -1.0236   | 0.1774 | -2.2450 | 0.1979  | False  |
| 5 | 6              | -0.6861   | 0.9811 | -2.8864 | 1.5142  | False  |
| 5 | 7              | 0.6718    | 0.9888 | -1.6847 | 3.0283  | False  |
| 5 | 8              | 0.6585    | 0.9869 | -1.5888 | 2.9059  | False  |
| 5 | 9+             | -0.5527   | 0.8731 | -1.7844 | 0.6790  | False  |
| 6 | 7              | 1.3580    | 0.8205 | -1.4405 | 4.1564  | False  |
| 6 | 8              | 1.3447    | 0.8020 | -1.3625 | 4.0519  | False  |
| 6 | 9+             | 0.1334    | 1.0000 | -1.8148 | 2.0816  | False  |
| 7 | 8              | -0.0133   | 1.0000 | -2.8489 | 2.8223  | False  |
| 7 | 9+             | -1.2246   | 0.6518 | -3.3476 | 0.8985  | False  |
| 8 | 9+             | -1.2113   | 0.5927 | -3.2124 | 0.7899  | False  |

Author contributions. PW developed and drafted this study and its data processing and analysis workflow (with inputs from the other authors), and implemented the workflow in Python. UAV aerial images of the Blueberry Hills region were acquired by students of Moose Kerr School Aklavik with OF, SK, MM, and CT. MM and CT provided a UAV flight design and generated the orthophoto from captured UAV images. The method for crowd-sourced mapping of ice-wedge polygon centroids was developed by OF and SM. Volunteered geographic information was collected at mapping events organized by OF, JL, ML, MM, SM, and PW. MM provided the validation of the Voronoi network. RF contributed to applications. ML, CT, and AZ served as principal investigators in the citizen science project. All authors contributed to editing and revising the manuscript.

#### Competing interests.

The authors declare no competing interests.

Acknowledgements. The UndercoverEisAgenten project was funded by the German Federal Ministry of Education and Research (German: Bundesministerium für Bildung und Forschung, BMBF) under the funding code 01BF2115C of the second Citizen Science funding guideline (2021–2024) (German: zweite Förderrichtlinie Citizen Science (2021–2024)). We would like to thank the volunteers participating in UAV image acquisition campaigns and in mapping sessions as a part of the project. The AWI Polar-5/6 research airplane was used to acquire MACS imagery of the Baldwin Peninsula, Alaska, during Perma-X campaigns. ML and SK acknowledge financial support from EU Horizon Europe under grant agreement No. 101133587 (ILLUQ). OF and SM acknowledge support by the Klaus Tschira Stiftung.

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
