# Peer review of "Monitoring Arctic Permafrost – Examining the Contribution of Volunteered Geographic Information to Mapping Ice-Wedge Polygons"

_EGUsphere, 2025_

## Author Comment (AC1)

We are grateful to the reviewers who suggested many helpful changes. They also made us aware of the parts that needed adaptation to ensure our concept was understandable to the reader. We considered all comments and our answers can be found in the following. The review comments are marked in bold and our answers in italic and blue colored font.

*Response to General comments (Review 1):*

**However, the discussion remains a bit superficial at some points and should be revised, for example, the broader relevance of the findings should be discussed more thoroughly.**

*All proposed inputs will be incorporated into the revised discussion.*

*Response to Specific comments (Review 1):*

**Line 13: Referring permafrost as a key climate variable should be considered again. I do not consider permafrost as a climate variable as its formation depends on the climate (and soil characteristics, moisture etc.) rather than it being a climate variable itself. Define permafrost properly.**

*While permafrost is listed by the World Meteorological Organization as an essential climate variable, we recognise that it is not a climate variable in the stricter physical sense. We agree to rephrase this.*

**Line 125: Is it possible to provide more information on how much smaller the area is if the volunteers are asked to trace the polygon outlines compared to just point digitization of the polygon centroids? This could help future researchers to plan better the best use of VGI for their own studies.**

*The exact difference in area that can be mapped is hard to quantify, as it not only depends on the mapping speed, but also on the volunteers' experience with the mapping project and the resulting willingness to contribute. As initial trials showed that volunteers reported challenges with outline digitisation of ice-wedge polygons that substantially reduced their motivation, we did not further pursue a systematic comparison of point vs polygon digitisation.*

**Clarify the use of different terms throughout the manuscript or use the same term consistently. In line 141 and in Fig. 1 in the first boxes from left, authors use the term "permafrost centroid", whereas the terms "ice-wedge centroids" or "ice-wedge polygon centroids" are used. Be consistent to avoid confusion.**

*We apologize for the confusion. We decided to use the term "ice-wedge polygon centroid" in the whole manuscript.*

**Post-hoc test (Tukey HSD) was not mentioned in the methods section, but first time mentioned in Appendix C, when presenting the results of the test. Please mention this also in the methods section.**

*We will introduce and reference Tukey HSD in the methods section.*

**Lines 167-169: Please clarify. Now it is unclear for the reader what does the "()rettelbach2021quantitative" refer to.**

*This was a formatting error of the citation, so we will correct it.*

**Lines 176-178: Clarify: In which one of these mapping events were the majority of the points digitized? Or were there three events organized simultaneously in the**

three different schools and one university, and majority of the points were digitized during one of these events organized at the same time at multiple locations?

*Most points were contributed by students from Higher Secondary Education (Gymnasium). The individual events resulted in different numbers of digitized points. As the events were organised one after the other at different locations, the exact format of each mapping event was slightly different.*

Line 197: Provide the number of the centroids also for BH as they were provided for the CB couple of lines earlier.

*We will do that.*

Lines: 207-208: Double check the distances, such figures are not presented in Table C1. This causes confusion, so please recheck, clarify, and/or revise Table C1.

*We will do that.*

Fig 6. and its caption: Add legend for the colors in the table or remove different colors if they are not necessary. Now the reader does not understand why the cells are in different colors and it is confusing.

*We will remove the color scale to avoid confusion.*

Major comment #1: The structure of the manuscript could be enhanced as some of the text is currently misplaced. For example, the section "4 Results" contain also discussion of the results (e.g. lines 201-202, 216-219, 229-237, and 280-284). Authors should either combine sections 4 and 5 into one section "4 Results and discussion" or then the current Discussion section should be expanded to contain all discussion currently found along the results. In its current form the Discussion section is quite superficial and short. In addition, the results section contains also some description of the methods (e.g. lines 239-248). There is also some overlap between results and methods (compare e.g. lines 162-163 and lines 260-263). Provide all needed info in the methods section.

*We have decided to better separate the Results and Discussion sections. All descriptive parts in the results will be removed and put into the methods section. The overlap between results and methods will also be taken out. All proposed inputs will be incorporated into the revised discussion.*

Major comment #2: How does the experience of the volunteers affect the results? In the study volunteers ranged from 12-18 years old to university students of geography. It can be assumed that geography students would have much more experience and knowledge on permafrost, the ice-wedge polygons, and GIS data compared to younger students. Were the tailored teaching materials presented before the mapping events enough to balance the different skill levels of the volunteers? It would be interesting to see separately the accuracy of the different volunteer groups compared to the experts. This info could help researchers plan such mapping events in the future to get as accurate data as possible. This would also give more content to the discussion of the limitations and potentials of the VGI in permafrost monitoring. Authors could at least discuss the potential influence of having volunteers with different experiences to the accuracy of the data.

*The application for crowd-sourced mapping of ice-wedge polygons was designed to enable contributions by people without any particular expertise – specifically: secondary school students as the main target group of the project – with some information and context provided through teaching materials. While prior knowledge allowed for providing more in-depth contextualisation to geography students at the respective events, the concrete*

*mapping task was designed to depend on general human pattern recognition skills. The majority of results were indeed generated by secondary school students. We do not provide a comparison of results from geography students and secondary school students from a controlled experimental setting.*

**Another possible topic to discuss is the different ways of acquiring remote sensing data. In CB aircraft data with resolution of 20 cm was utilized, whereas in BH drone data with ~10 cm resolution was collected. Does different RS data affect the mapping results? Which alternative is preferrable in future studies if one can choose?**

*While we did not experimentally compare the mapping results from 20 cm and 10 cm resolution images, we do not expect resolution to be a significant factor determining the results at this level. In contrast, mapping results are impacted by other aspects of image quality such as lighting, contrast, blurriness, and distortions, as stated in 5.2.*

*We did in fact deliberately downsample the drone image based orthomosaic to 10 cm for practical reasons, as we deemed the native ~5 cm resolution excessive for our use case. However, given the common size of ice-wedge polygons and the width of their delimiting troughs, crowd-sourced mapping of these features certainly relies on high-resolution imagery. We will add this to the discussion section.*

**In addition, the applicability of the method for monitoring other permafrost landforms could be discussed shortly to provide broader perspectives of the potential/limitations of the VGI in permafrost monitoring.**

*Thanks for the suggestion, we will add this aspect to the discussion. Our proposed method for network reconstruction from crowd-sourced centerpoints is specifically adapted to monitoring ice-wedge polygons. Crowd-sourced mapping with micro-tasking in general, however, is an approach potentially useful in various applications of environmental monitoring (not exclusively) of other permafrost landforms such as pingos, thaw slumps, etc., provided these landforms are visible in the available imagery to the non-expert eye. With regard to volunteer engagement, one limitation consists in the requirement that the feature of interest should not be too sparsely distributed within the area of the crowd-sourced mapping exercise.*

**Line 325-326: Was this shown in this study or elsewhere? Please add references if needed**

*These are only indicative and unpublished results from early trials conducted prior to this study to compare different task designs.*

**Lines 329-331: Previously it was stated (in lines 216-219) that this result is in line with previous research (Herfort, 2018). Please add references also here. In general, there are relatively few references in the Discussion section, incorporating more references into the discussion could enhance its quality.**

*We will add the reference to the discussion session in the revised manuscript.*

*Response to Technical suggestions (Review 1):*

All of the following technical suggestions will be fixed as proposed by the reviewer:

**Lines 13-21: Could these paragraphs be combined? Now the first paragraph is very short, which hampers the flow of the text.**

**Caption for Fig. 1: The alphabets referring to the subfigures are messed up ("c" is**

marked twice in the caption, the latter one should be "d" and current "d" should be "e").

Line 112: correct "(see A)" to form "(see Appendix A)"

Line 127: add word "Sect." to "described in 3.3". The instructions of TC state: "The abbreviation "Sect." should be used when it appears in running text and should be followed by a number unless it comes at the beginning of a sentence."

Line 132: Same here: "(see 3.3)" à "(see Sect. 3.3.)"

Lines 140-149: Could these paragraphs be combined? Now the first paragraph is very short, which hampers the flow of the text.

Line 190: Here the "Section" is used. However, correct to: "(see Sect. 3.2)"

Captions for Figs. 5&6: Be consistent, if you explain the outliers in caption of Fig. 6, explain them also in caption of Fig. 5.

Fig 6., Table 1, Fig. 9.: To get a more polished outlook replace underscores with spaces. In the caption for Table 1 the abbreviations for study regions are provided but the abbreviations are not used in the table à these could be removed in the name of the consistency (abbreviations are not presented in other captions).

There are lots of good figures in the manuscript. However, authors could consider combining similar figures together to facilitate comparisons between the research areas. For example, figures 3&4, 7&8, and 10&11 could be combined into subfigures.

*Agreed, we will combine figures 3 & 4, 7 & 8, and 10 & 11 into subfigures.*

Line 273-273: correct "(described in section 4.4)" to form "(described in Sect. 4.4).

Line 274: remove extra parentheses after Appendix B.

First paragraph of the section 5.2.: The paragraph is quite long, and it could be divided into two after the sentence: "In this study, we identify a number of five to six contributions per task as sufficient for assuring a high quality of aggregated crowd-source data."

Line 347: Remove "(Volunteered Geographic Information)". It is unnecessary at this point of the manuscript.

Last paragraph of the section 5.2.: Quite long paragraph again, which could be divided into two to enhance the readability of it. Division could be made for example after sentence "It has been demonstrated that these types of polygons can be well-represented by Voronoi tessellations (Cresto Aleina et al., 2013; Ulrich et al., 2014)."

Appendix A: Short paragraphs hamper the flow of the text. All paragraphs could be combined into one.

Appendix B: Very small detail, but in general the results are presented first for the CB and then to BH study region throughout the manuscript. However, in Appendix B, results are presented first for the BH and then for CB. It would be more consistent to keep the same order if there is no specific reason to change it.

---

## Author Comment (AC2)

We are grateful to the reviewers who suggested many helpful changes. They also made us aware of the parts that needed adaptation to ensure our concept was understandable to the reader. We considered all comments and our answers can be found in the following. The review comments are marked in bold and our answers in italic and blue colored font.

*Response to General comments (Review 2):*

Although technical details of the crow-sourced mapping application are available in Appendix A, however, if some details and potential issues could be mentioned or discussed in the manuscript, it would benefit other crow-sourced applications. A fundamental assumption of using a crow-sourced system is that it allows many people to contribute to a task that cannot be completed by a few experts. If the task does attract many people, how many concurrent users are allowed in the system? How to synchronize data? In fact, it's not easy to recruit volunteers for mapping specific features (e.g., permafrost-thaw features) that most people are not familiar with, as demonstrated by Huang et al. 2023 (Huang, L., et al. Identifying active retrogressive thaw slumps from ArcticDEM. ISPRS Journal of Photogrammetry and Remote Sensing, 205, 301–316), how are you going to recruit contributors for a continuous monitoring task besides mapping events? How to manage large datasets if you use the crow-sourced system for much larger areas?

*Thanks for the suggestion, we will gladly provide more detail on the mapping application, such as its task assignment strategies and data management, in the appendix. With regard to the issue of recruiting contributors, we will add some ideas to the discussion section, e.g. the integration with popular crowd-sourced mapping apps, and the continuous collaboration with educational institutions.*

*Response to Specific comments (Review 2):*

**L6: "positional accuracies", validated against what data?**

*This was validated against polygon center points mapped by experts. We will state this more explicitly in the revised manuscript.*

**L13: "the largest non-seasonal component of the cryosphere"? largest in area?**

It is the largest in area indeed. We will state this more explicitly.

**A screenshot of the web-based crow-sourced mapping application would be helpful for readers to understand its functions and capabilities.**

*Thanks for the suggestion, we will add a screenshot of the application to the respective section of the annex.*

**L146: "crowd-validated"? I am a little confused, as these results will still need to be validated by the experts?**

*We consider the centers of the clustered volunteer-contributed ice-wedge polygon centroids as "crowd-validated", but we agree to remove this term and replace it with "volunteer-contributed" as it might be ambiguous.*

**Figure 3 would be good to show a zoom-in region, like Figure 4.**

*We add the zoom-in region to the figure.*

**Figure 7, please show a zoom-in Figure, like Figure 8.**

*We add the zoom-in region to the figure.*

**L239: Where is the difference between "manually digitized reference polygons" and "expert-derived polygons"?**

Expert-derived polygons are generated via the network reconstruction method described in the manuscript from approximate polygon **center points** digitized by experts. Reference polygons are manually digitized **as polygons** by experts without the need to fall back on the network reconstruction method. Comparing expert-derived polygons with reference polygons manually digitized by experts allows for assessing the quality of the output of the network reconstruction method.

**L286: "betweenness" a sentence to explain betweenness and its importance would be helpful for readers without a hydrological background.**

Betweenness centrality provides a measure of the importance of individual channels for water drainage within hydrological networks (Marra et al., 2021). Channels with high centrality act as critical connectors, linking otherwise isolated parts of the network and thereby playing a key role in maintaining or enabling overall drainage. In the context of the hydrological function of ice-wedge polygon networks, through segments with high centrality are likely to carry disproportionately large water fluxes, as they concentrate flow. Consequently, they play an important role in the transport of dissolved nutrients and other substances, while also being more susceptible to enhanced erosion and thermokarst development (Rettelbach et al., 2021).

**L324: "the overall time available for the crowd-sourced mapping process", What's the time referring to? The event duration?**

*This does not necessarily refer to a single event, but to the overall person-hours of volunteer contributions that can be mobilised for a specific crowd-sourced mapping process, i.e. the mapping of a given area of interest. We will reformulate in the manuscript to clarify.*

**L369: "especially when high-resolution elevation data is unavailable"? This is confusing. This manuscript still requires high-resolution imagery. From my understanding, the need for spatial resolution is determined by the observing objects, that is, smaller features require higher spatial resolution.**

*"High-resolution" here refers to horizontal resolution and vertical accuracy of elevation data. Automated processes often depend on high resolution elevation data, e.g. to detect subtle elevation differences in narrow ice-wedge polygon rims/troughs. Elevation data of the necessary resolution is not globally available. Our approach does not depend on elevation data at all, but it does require high-resolution imagery.*

*Response to Technical suggestions (Review 2):*

**L169: "()rettel-bach2021quantitative"?**

*This is a formatting error of the citation to be corrected.*

**L309: change "inSAR" to "InSAR".**

*This is to be corrected.*

References:

Marra, W.A.; Kleinhans, M.G.; Addink, E.A. Network concepts to describe channel importance and change in multichannel systems: Test results for the Jamuna River, Bangladesh. Earth Surf. Process. Landforms 2014, 39, 766–778.

Rettelbach, T.; Langer, M.; Nitze, I.; Jones, B.; Helm, V.; Freytag, J.-C.; Grosse, G. A Quantitative Graph-Based Approach to Monitoring Ice-Wedge Trough Dynamics in Polygonal Permafrost Landscapes. Remote Sens. 2021, 13, 3098. https://doi.org/10.3390/rs13163098